# Rethinking Tokenized Graph Transformers for Node Classification

**Jinsong Chen**[1,2]* **Chenyang Li**[1,3]* **Gaichao Li**[1,3]**, John E. Hopcroft**[3,4]**, Kun He**[1,3]†

[1]School of Computer Science and Technology, Huazhong University of Science and Technology
[2]Faculty of Artificial Intelligence in Education, Central China Normal University
[3]Hopcroft Center on Computing Science, Huazhong University of Science and Technology
[4]Department of Computer Science, Cornell University
guangnianchenai@ccnu.edu.cn, {chenyangli,gaichaolee}@hust.edu.cn,
jeh@cs.cornell.edu, brooklet60@hust.edu.cn

## Abstract

Node tokenized graph Transformers (GTs) have shown promising performance in node classification. The generation of token sequences is the key module in existing tokenized GTs which transforms the input graph into token sequences, facilitating the node representation learning via Transformer. In this paper, we observe that the generations of token sequences in existing GTs only focus on the first-order neighbors on the constructed similarity graphs, which leads to the limited usage of nodes to generate diverse token sequences, further restricting the potential of tokenized GTs for node classification. To this end, we propose a new method termed SwapGT. SwapGT first introduces a novel token swapping operation based on the characteristics of token sequences that fully leverages the semantic relevance of nodes to generate more informative token sequences. Then, SwapGT leverages a Transformer-based backbone to learn node representations from the generated token sequences. Moreover, SwapGT develops a center alignment loss to constrain the representation learning from multiple token sequences, further enhancing the model performance. Extensive empirical results on various datasets showcase the superiority of SwapGT for node classification. Code is available at https://github.com/JHL-HUST/SwapGT.

## 1 Introduction

Node classification [22], the task of predicting node labels in a graph, is a fundamental problem in graph data mining with numerous real-world applications. Graph Neural Networks (GNNs) [42, 23] have traditionally been the dominant approaches. However, the message passing mechanism inherent to GNNs suffers from some limitations, such as over-smoothing [5], which prevents them from effectively capturing deep graph structural information and hinders their performance in downstream tasks. In contrast, Graph Transformers (GTs), which adapt the Transformer framework for graph-based learning, have emerged as a promising alternative, demonstrating impressive performance in node classification. Existing GTs can be broadly classified into two categories based on their model architecture: hybrid GTs and tokenized GTs.

Hybrid GTs combine the strengths of GNN and Transformer, using GNNs to capture local graph topology and Transformers to model global semantic relationships. However, recent studies have highlighted a key issue with this approach: directly modeling semantic correlations among all node pairs using Transformers can lead to the over-globalization problem [39], which compromises model

---

*Equal contribution.
†Corresponding author.

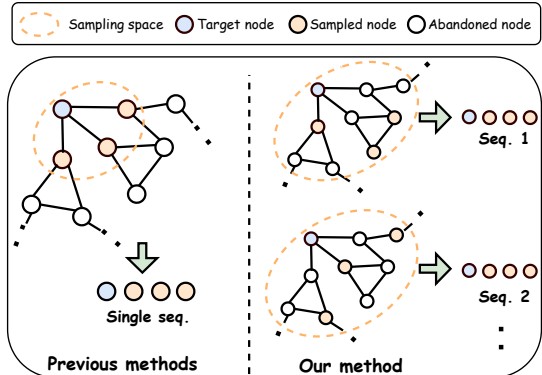

Figure 1: The toy example of token generation on the constructed $k$-NN graph. Previous methods only focus on 1-hop neighborhood to construct a single token sequence. While our method can flexibly select tokens from multi-hop neighborhoods to generate diverse token sequences.

performance. Tokenized GTs, on the other hand, generate independent token sequences for each node, which encapsulate both local topological and global semantic information. Transformer models are then utilized to learn node representations from these token sequences. The advantage of tokenized GTs is that they limit the token sequences to a small, manageable number of tokens, naturally avoiding the over-globalization issue. In tokenized GTs, the token sequences typically include two types of tokens: neighborhood tokens and node tokens. Neighborhood tokens aggregate multi-hop neighborhood information of a target node, while node tokens are sampled based on the similarity between nodes.

However, recent studies [17] have shown that neighborhood tokens often fail to preserve complex graph properties such as long-range dependencies and heterophily, limiting the richness of node representations. On the other hand, node tokens, generated through various sampling strategies, can better capture correlations between nodes in both feature and topological spaces [44, 7], making them more effective in preserving complex graph information. As a result, this paper focuses on node token-based GTs.

A recent study [7] formalized the node token generation process as two key steps: similarity evaluation and top-$k$ sampling. In the first step, similarity scores between node pairs are calculated based on different similarity measures to preserve the relations of nodes in different feature spaces. While in the second step, the top $k$ nodes with the highest similarity scores are selected as node tokens to construct the token sequence. In this paper, we provide a new perspective on token generation in existing tokenized GTs. We identify that the token generation process can be viewed as a neighbor selection operation on the $k$-nearest neighbor ($k$-NN) graph. Specifically, a $k$-NN graph is constructed based on node pair similarities, and the neighbor nodes within the first-order neighborhood of each node are selected to form the token sequence.

Figure 1 illustrates this idea with a toy example. We can observe that only a small subset of nodes is selected via existing token generation strategies, which indicates that existing methods have limited exploitation of the $k$-NN graph and are unable to comprehensively utilize the correlations between node pairs to explore more informative nodes with potential association to construct token sequences. This situation inevitably restricts the ability of tokenized GTs to capture informative node representations. Furthermore, in scenarios with sparse training data, relying on token sequences generated from a limited set of nodes may lead to over-fitting, as Transformers, being complex models, may struggle to generalize effectively.

This leads to the following research question: *How can we more comprehensively and effectively exploit node pair correlations to generate diverse token sequences, thus improving the performance of tokenized GTs for node classification?* To address this, we introduce a novel method called SwapGT. Specifically, SwapGT introduces a new operation, token swapping, which leverages the semantic correlations of nodes in the $k$-NN to swap tokens in different token sequences, generating more diverse token sequences. By incorporating multiple token sequences, SwapGT enables the model to

learn more comprehensive node representations. Additionally, SwapGT employs a Transformer-based backbone and introduces a tailored readout function to learn node representations from the generated token sequences. To handle the case where a node is assigned multiple token sequences, we propose a center alignment loss to guide the training process. The main contributions of this paper are summarized as follows:

- We propose a novel token swapping operation that fully exploits semantic correlations of nodes to generate diverse token sequences.

- We develop a Transformer-based backbone with a center alignment loss to learn node representations from the generated diverse token sequences.

- Extensive experiments on various datasets with different training data ratios showcase the effectiveness of SwapGT in node classification.

## 2 Relation Work

### 2.1 Graph Neural Networks

GNNs [41, 43, 19, 9, 8, 21] have shown remarkable performance in this task. Previous studies [33, 40, 36, 24] have primarily concentrated on the incorporation of diverse graph structural information into the message-passing framework. Classic deep learning techniques, such as the attention mechanism [33, 4] and residual connections [40, 14], have been exploited to enhance the information aggregation on graphs. Moreover, aggregating information from high-order neighbors [24, 1, 46, 47] or nodes with high similarity across different feature spaces [30] has been demonstrated to be efficacious in improving model performance. Follow-up GNNs have focused on the utilization of complex graph features to extract distinctive node representations. A prevalent strategy entails the utilization of signed aggregation weights [3, 15, 25, 20, 13] to optimize the aggregation operation. In addition, Zhang *et al*. [42] have excellently integrated LLMs with GNNs, employing causal inference to mitigate biases introduced by the LLMs. Nevertheless, restricted by the inherent limitations of message-passing mechanism, the potential of GNNs for graph data mining has been inevitably weakened. Developing a new graph deep learning paradigm has attracted great attention in graph representation learning.

### 2.2 Graph Transformers

GTs [16, 28, 39, 48] have emerged as a novel architecture for graph representation learning and have exhibited substantial potential in node classification. A commonly adopted design paradigm for GTs is the combination of Transformer modules with GNN-style modules to construct hybrid neural network layers, called hybrid GTs [37, 38, 2]. In this design, Transformer is employed to capture global information, while GNNs are utilized for local information extraction [32, 16, 10]. Despite effectiveness, directly utilizing Transformer to model the interactions of all node pairs could occur the over-globalization issue [39], inevitably weakening the potential for graph representation learning.

An alternative yet effective design of GTs involves transforming the input graph into independent token sequences termed tokenized GTs [44, 6, 17, 27, 7, 12], which are then fed into the Transformer layer for node representation learning. Neighborhood tokens [6, 11, 27, 17, 7] and node tokens [44, 7, 17] are two typical elements in existing tokenized GTs. The former, generally constructed by propagation approaches, such as random walk [6, 11] and personalized PageRank [17]. The latter is generated by diverse sampling methods based different similarity measurements, such as PageRank score [17] and attribute similarity [44]. Since tokenized GTs only focus on the generated tokens, they naturally avoiding the over-globalization issue.

As pointed out in previous study [17], node token oriented GTs are more efficient in capturing various graph information, such as long-range dependencies and heterophily, compared to neighborhood token oriented GTs. However, we identify that previous methods only leverage a small subset of nodes as tokens for node representation learning, which could limit the model ability of deeply exploring graph information. In this paper, we develop a new method SwapGT that introduces a novel token swapping operation to produce more informative token sequences, further enhancing the model performance.

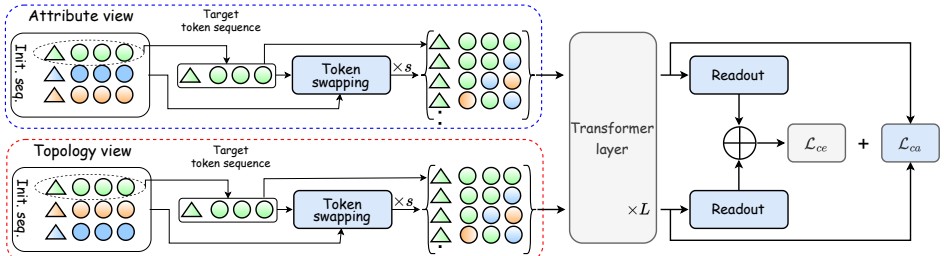

Figure 2: The overall framework of SwapGT. First, we generate the initial token sequences from both the attribute view and topology view. Then, we utilize the proposed token swapping operation to generate new token sequences for each target node. These generated token sequences are then fed into a Transformer-based backbone to learn node representations and generate predicted labels. Additionally, a center alignment loss is adopted to further constrain the representations extracted from different token sequences.

## 3 Preliminaries

Suppose an attributed graph is denoted as $\mathcal{G} = (V, E, \mathbf{X})$ where $V$ and $E$ are the sets of nodes and edges in the graph. $\mathbf{X} \in \mathbb{R}^{n \times d}$ is the attribute feature matrix, where $n$ and $d$ are the number of nodes and the dimension of the attribute feature vector, respectively. We also have the adjacency matrix $\mathbf{A} = \{0, 1\}^{n \times n}$. If there is an edge between nodes $v_i$ and $v_j$, $\mathbf{A}_{ij} = 1$; otherwise, $\mathbf{A}_{ij} = 0$. $\hat{\mathbf{A}}$ denotes the normalized version calculated as $\hat{\mathbf{A}} = (\mathbf{D} + \mathbf{I})^{-1/2}(\mathbf{A} + \mathbf{I})(\mathbf{D} + \mathbf{I})^{-1/2}$ where $\mathbf{D}$ and $\mathbf{I}$ are the diagonal degree matrix and the identity matrix, respectively. In the scenario of node classification, each node is associated with a one-hot vector to identify the unique label information, resulting in a label matrix $\mathbf{Y} \in \mathbb{R}^{n \times c}$ where $c$ is the number of labels. Given a set of labeled nodes $V_L$, the goal of the task is to predict the labels of the rest nodes in $V - V_L$.

## 4 Methodology

In this section, we detail our proposed SwapGT, involving two main stages: token sequences generation via token swapping and token representation learning with Transformer-based backbone. The overall framework of SwapGT is shown in Figure 2.

### 4.1 Token Sampling

Token sampling, which aims to select relevant nodes from input graph to construct a token sequence for each target node, is a crucial operation in recent GTs [44, 17]. Generally speaking, the token sampling operation for the target node $v_i$ can be summarized as follows:

$$N_i = \{v_j | v_j \in \text{Top}(\mathbf{S}_i, k)\}, \tag{1}$$

where $N_i$ denotes the token set of node $v_i$. $\text{Top}(\cdot)$ denotes the top-$k$ sampling operation and $k$ is the number of sampled tokens. $\mathbf{S}_i \in \mathbb{R}^{1 \times n}$ denotes the vector of similarity scores between $v_i$ and other nodes. Obviously, different strategies for measuring node similarity will result in different sets of sampled tokens. According to empirical results in previous studies [44, 17], measuring the node similarity from different feature spaces to sample different types of tokens can effectively enhance the model performance. Hence, in this paper, we sample the tokens from both the attribute feature space and the topology feature space, resulting in two token sets $N_i^A$ and $N_i^T$, respectively.

Specifically, for $N_i^A$, we utilize the raw attribute features to calculate the cosine similarity of each node pair to obtain the similarity scores. For $N_i^T$, we adopt the neighborhood features to represent the topology features of nodes. The neighborhood feature matrix is calculated as $\mathbf{X}' = \phi(\hat{\mathbf{A}}, \mathbf{X}, K)$ where $K$ denotes the number of propagation steps and $\phi(\cdot)$ denotes the personalized PageRank propagation [24]. The influence of $K$ on the model performance is discussed in Appendix A.1. Then we leverage the neighborhood features to measure the similarity of nodes in the topology feature space via the cosine similarity.

**Algorithm 1** The Token Swapping Algorithm

---

**Input:** Sampled token set of all nodes $N \in \mathbb{R}^{n \times k}$; Target node $v_i$; Probability $p$; Swapping times $t$
**Output:** The new token set $N_i' \in \mathbb{R}^{1 \times k}$ of $v_i$
1: Initialize $N_i' = N_i$;
2: **for** $t_0 = 1$ **to** $t$ **do**
3:     Initialize $N^{new} = \{\}$;
4:     **for** $v_j \in N_i'$ **do**
5:         **if** $random(0,1) > p$ **then**
6:             $N^{new} = N^{new} \cup \{v_j\}$;
7:         **else**
8:             $v^{new} = \zeta(N_j)$;
9:             $N^{new} = N^{new} \cup \{v^{new}\}$;
10:         **end if**
11:     **end for**
12:     $N_i' = N^{new}$;
13: **end for**
14: **return** $N_i'$

---

## 4.2 Token Swapping

As discussed in Figure 1, existing node token generators [45, 44, 17] could be regarded as selecting the 1-hop neighborhood nodes in the constructed $k$-NN graph, which is inefficient in fully leveraging the semantic relevance between nodes and restricts the diversity of the token sequences, further limiting the model performance. To effectively obtain diverse token sequences, SwapGT introduces a novel operation based on the unique characteristic of token sequences, called token swapping. The key idea is to swap tokens in different sequences to construct new token sequences. Specifically, for each node token $v_j$ in the token set of node $v_i$, we generate a new node token $v^{new} = \zeta(N_j)$ based on the token set $N_j$ of $v_j$, where $\zeta(\cdot)$ denotes the random sampling operation. Then, the new token set $N_i'$ of $v_i$ is generated as follows:

$$N_i' = \{\zeta(N_j)|v_j \in N_i\}. \tag{2}$$

Equation 2 indicates that for each node token $v_j$, we swap it with a random node $v^{new}$ in its node token set $N_j$ to construct the new token set $N_i'$ for the target node $v_i$. Appendix B provides a toy example to illustrate the token swapping operation.

Here, we provide deep insights for the token swapping operation. According to Figure 1, nodes in the token set are the 1-hop neighbors of the target node in the $k$-NN graph. Therefore, the selected node $v^{new}$ is within the 2-hop neighborhoods of $v_i$. Therefore, performing the swapping operation $t$ times is equal to enlarge the sampling space from 1-hop neighborhood to $(t+1)$-hop neighborhood. Hence, the swapping operation effectively utilizes the semantic relevance between nodes to generate diverse token sequences. The overall algorithm of token swapping is summarized in Algorithm 1. Moreover, we theoretically analyze the probability of sampling neighbors from different $k$-NN hops on the constructed $k$-NN graph:

**Fact 1.** *If we guarantee that $(t+1)p < 1$, in the generated token sequences, the probability of sampling lower-order neighbors is greater than sampling higher-order neighbors on the constructed $k$-NN graph, where $t$ denotes the maximum swapping times, and $p$ denotes the probability of swapping tokens per time in Algorithm 1.*

The detailed proof of **Fact 1** is provided in Appendix C. **Fact 1** indicates that SwapGT prefers to sample low-order neighbors on the constructed $k$-NN graph which are more likely to be relevant, reducing the potential noise and effectively generating relevant token sequences for each node.

After generating various token sets, we utilize them to construct the input token sequence for representation learning. Given a token set $N_i$, the input token sequence $\mathbf{Z}_i^{in}$ of node $v_i$ is as:

$$\mathbf{Z}_i^{in} = [\mathbf{X}_i, \mathbf{X}_{N_{i,0}}, \ldots, \mathbf{X}_{N_{i,k-1}}]. \tag{3}$$

By performing Algorithm 1 $s$ times, we can finally obtain $1 + s$ token sets for the target node $v_i$. We combine all token sequences generated by Equation 3 based on different token sets to obtain the final

input token sequences $\mathbf{Z}_i \in \mathbb{R}^{(1+s)\times(1+k)\times d}$. Following the same process, the input token sequences generated by $N_i^A$ and $N_i^T$ are denoted as $\mathbf{Z}_i^A$ and $\mathbf{Z}_i^T$, respectively.

## 4.3  Transformer-based Backbone

The proposed Transformer-based backbone aims to learn node representations and predict the node labels according to the input token sequences. Take the input sequence $\mathbf{Z}_i^A$ for example, we first utilize the projection layer to obtain the model input:

$$\mathbf{Z}_i^{A,(0)} = \rho(\mathbf{Z}_i^A), \tag{4}$$

where $\rho(\cdot)$ denotes the projection layer and $\mathbf{Z}_i^{A,(0)} \in \mathbb{R}^{(1+s)\times(k+1)\times d_0}$ denotes the model input of node $v_i$. Then, a Transformer layer-based encoder is applied to learn node representations from the model input:

$$\mathbf{Z}_i^{\prime A,(l)} = \mathrm{MSA}\left(\mathrm{LN}\left(\mathbf{Z}_i^{A,(l-1)}\right)\right) + \mathbf{Z}_i^{A,(l-1)}, \tag{5}$$

$$\mathbf{Z}_i^{A,(l)} = \mathrm{FFN}\left(\mathrm{LN}\left(\mathbf{Z}_i^{\prime A,(l)}\right)\right) + \mathbf{Z}_i^{\prime A,(l)}, \tag{6}$$

where $l = 1, \ldots, L$ indicates the $l$-th Transformer layer. MSA,FFN and $\mathrm{LN}(\cdot)$ denote the multi-head self-attention mechanism, feed-forward networks and the LayerNorm operation in the standard Transformer layer.

Through the encoder, we obtain the representations of input token sequences $\mathbf{Z}_i^{A,(L)} \in \mathbb{R}^{(1+s)\times(1+k)\times d_L}$. Then, we take the representation of the first item in each token sequence as the final representation of the input sequence. This is because the first item in each token sequence is the target node itself, and the output representation has learned necessary information from other sampling tokens in the input sequence via the Transformer encoder.

Hence, the final output of the Transformer encoder is denoted as $\mathbf{Z}^{A,i} \in \mathbb{R}^{(1+s)\times d_L}$. Then, we utilize the following readout function to obtain the node representation learned from multi token sequences:

$$\mathbf{Z}_i^{A,F} = \mathbf{Z}_0^{A,i}||(\frac{1}{s}\sum_{j=1}^{s}\mathbf{Z}_j^{A,i}), \tag{7}$$

where $||$ denotes the concatenation operation, $\mathbf{Z}_i^{A,F} \in \mathbb{R}^{1\times d_L}$ denotes the representation of $v_i$ learned from token sequences generated from the attribute feature space. Similarly, we can obtain $\mathbf{Z}_i^T$ from the topology feature space. To effectively utilize information of different feature spaces, we leverage the following strategy to fuse the learned representations:

$$\mathbf{Z}_i^F = \alpha \cdot \mathbf{Z}_i^{A,F} + (1-\alpha) \cdot \mathbf{Z}_i^{A,T}, \tag{8}$$

where $\alpha \in [0,1]$ is a hyper-parameter to balance the contribution from attribute features and topology features on the final node representation. At the end, we adopt Multi-Layer Perception-based predictor for label prediction and the cross-entropy loss for model training:

$$\mathcal{L}_{ce} = -\sum_{i\in V_L}\sum_{j=0}^{c}\mathbf{Y}_{i,j}\ln\mathbf{Y}'_{i,j}, \mathbf{Y}'_i = \mathrm{MLP}(\mathbf{Z}_i^F). \tag{9}$$

## 4.4  Center Alignment Loss

To further enhance the model's generalization, we develop a center alignment loss to constrain the representations learned from different token sequences for each node. Specifically, given the representations of multi-token sequences $\mathbf{Z}^i \in \mathbb{R}^{(1+s)\times d_L}$, we first calculate the center representation $\mathbf{Z}_c^i = \frac{1}{(1+s)}\sum_{j=0}^{s}\mathbf{Z}_j^i$. Then, the center alignment loss is calculated as follows:

$$\mathrm{CAL}(\mathbf{Z}^i, \mathbf{Z}_c^i) = 1 - \frac{1}{(1+s)}\sum_{j=0}^{s}\mathrm{Cosine}(\mathbf{Z}_j^i, \mathbf{Z}_c^i), \tag{10}$$

Table 1: Comparison of all models in terms of mean accuracy $\pm$ stdev (%) under dense splitting. The best results appear in **bold**. The second results appear in underline.

| Dataset | Photo | ACM | Computer | Citeseer | WikiCS | BlogCatalog | UAI2010 | Flickr |
|---|---|---|---|---|---|---|---|---|
| $\mathcal{H}$ | 0.83 | 0.82 | 0.78 | 0.74 | 0.66 | 0.40 | 0.36 | 0.24 |
| SGC | $93.74_{\pm0.07}$ | $93.24_{\pm0.49}$ | $88.90_{\pm0.11}$ | $76.81_{\pm0.26}$ | $76.67_{\pm0.19}$ | $72.61_{\pm0.07}$ | $69.87_{\pm0.17}$ | $47.48_{\pm0.40}$ |
| APPNP | $94.98_{\pm0.41}$ | $93.00_{\pm0.55}$ | $91.31_{\pm0.29}$ | $77.52_{\pm0.22}$ | $81.96_{\pm0.14}$ | $94.77_{\pm0.19}$ | $77.41_{\pm0.47}$ | $84.66_{\pm0.31}$ |
| GPRGNN | $94.57_{\pm0.44}$ | $93.42_{\pm0.20}$ | $90.15_{\pm0.34}$ | $77.59_{\pm0.36}$ | $82.43_{\pm0.29}$ | $94.36_{\pm0.29}$ | $76.94_{\pm0.64}$ | $85.91_{\pm0.51}$ |
| FAGCN | $94.06_{\pm0.03}$ | $93.37_{\pm0.24}$ | $83.17_{\pm1.81}$ | $76.19_{\pm0.62}$ | $79.89_{\pm0.93}$ | $79.92_{\pm4.39}$ | $72.17_{\pm1.57}$ | $82.03_{\pm0.40}$ |
| BM-GCN | $95.10_{\pm0.20}$ | $93.68_{\pm0.34}$ | $91.28_{\pm0.96}$ | $77.91_{\pm0.58}$ | $83.90_{\pm0.41}$ | $94.85_{\pm0.42}$ | $77.39_{\pm1.13}$ | $83.97_{\pm0.87}$ |
| ACM-GCN | $94.56_{\pm0.21}$ | $93.04_{\pm1.28}$ | $85.19_{\pm2.26}$ | $77.62_{\pm0.81}$ | $83.95_{\pm0.41}$ | $94.53_{\pm0.53}$ | $76.87_{\pm1.42}$ | $83.85_{\pm0.73}$ |
| ANS-GT | $94.88_{\pm0.17}$ | $93.92_{\pm0.41}$ | $89.58_{\pm0.34}$ | $77.54_{\pm0.26}$ | $82.53_{\pm0.28}$ | $91.93_{\pm0.38}$ | $74.16_{\pm1.28}$ | $85.94_{\pm0.58}$ |
| NAGphormer | $95.47_{\pm0.29}$ | $93.32_{\pm0.30}$ | $90.79_{\pm0.45}$ | $77.68_{\pm0.73}$ | $83.61_{\pm0.28}$ | $94.42_{\pm0.63}$ | $76.36_{\pm1.12}$ | $86.85_{\pm0.85}$ |
| SGFormer | $92.93_{\pm0.12}$ | $93.79_{\pm0.34}$ | $81.86_{\pm3.82}$ | $77.86_{\pm0.76}$ | $79.65_{\pm0.31}$ | $94.33_{\pm0.19}$ | $57.98_{\pm3.95}$ | $61.05_{\pm0.68}$ |
| Specformer | $95.22_{\pm0.13}$ | $93.63_{\pm1.94}$ | $85.47_{\pm1.44}$ | $77.96_{\pm0.89}$ | $83.74_{\pm0.62}$ | $94.21_{\pm0.23}$ | $73.06_{\pm0.77}$ | $86.55_{\pm0.40}$ |
| VCR-Graphormer | $95.38_{\pm0.51}$ | $93.11_{\pm0.79}$ | $90.47_{\pm0.58}$ | $77.21_{\pm0.65}$ | $80.82_{\pm0.72}$ | $94.19_{\pm0.17}$ | $76.08_{\pm0.52}$ | $85.96_{\pm0.55}$ |
| CoBFormer | $95.29_{\pm0.35}$ | $93.82_{\pm0.58}$ | $90.21_{\pm0.89}$ | $77.74_{\pm0.72}$ | $83.28_{\pm0.68}$ | $93.98_{\pm0.72}$ | $76.85_{\pm0.69}$ | $86.84_{\pm0.78}$ |
| PolyFormer | $95.45_{\pm0.21}$ | $94.27_{\pm0.44}$ | $90.87_{\pm0.74}$ | $78.03_{\pm0.86}$ | $83.79_{\pm0.75}$ | $95.08_{\pm0.43}$ | $77.92_{\pm0.82}$ | $87.01_{\pm0.57}$ |
| SwapGT | $\mathbf{95.92_{\pm0.18}}$ | $\mathbf{94.98_{\pm0.41}}$ | $\mathbf{91.73_{\pm0.72}}$ | $\mathbf{78.49_{\pm0.95}}$ | $\mathbf{84.52_{\pm0.63}}$ | $\mathbf{95.93_{\pm0.56}}$ | $\mathbf{79.06_{\pm0.73}}$ | $\mathbf{87.56_{\pm0.61}}$ |

where $\mathrm{Cosine}(\cdot)$ denotes the cosine similarity function.

The rationale of Equation 10 is that the representations learned from different token sequences can be regarded as different views of the target node. Therefore, these representations should naturally be close to each other in the latent space. In practice, we separately calculate the center alignment loss of token sequences from different feature spaces:

$$\mathcal{L}_{ca} = \mathrm{CAL}(\mathbf{Z}^{A,i}, \mathbf{Z}_c^{A,i}) + \mathrm{CAL}(\mathbf{Z}^{T,i}, \mathbf{Z}_c^{T,i}). \tag{11}$$

The overall loss of SwapGT is as follows:

$$\mathcal{L} = \mathcal{L}_{ce} + \lambda \cdot \mathcal{L}_{ca}, \tag{12}$$

where $\lambda$ is a coefficient controlling the balance between the two loss functions.

## 5 Experiments

### 5.1 Dataset

We adopt eight widely used datasets, involving homophily and heterophily graphs: Photo [6], ACM [34], Computer [6], BlogCatalog [29], UAI2010 [35], Flickr [29] and Wiki-CS [31]. The edge homophily ratio [25] $H(\mathcal{G}) \in [0, 1]$ is adopted to evaluate the graph's homophily level. $H(\mathcal{G}) \to 1$ means strong homophily, while $H(\mathcal{G}) \to 0$ means strong heterophily. Statistics of datasets are summarized in Appendix D.1. To comprehensively evaluate the model performance in node classification, we provide two strategies to split datasets, called dense splitting and sparse splitting. In dense splitting, we randomly choose 50% of each label as the training set, 25% as the validation set, and the rest as the test set, which is a common setting is previous studies [37, 38]. While in sparse splitting [15], we adopt 2.5%/2.5%/95% splitting for training set, validation set and test set.

### 5.2 Baseline

We adopt 13 representative approaches as the baselines: SGC [36], APPNP [24], GPRGNN [15], FAGCN [3], BM-GCN [18], ACM-GCN [26], ANS-GT [44], NAGphormer [6], SGFormer [38], Specformer [2], VCR-Graphormer [17], CoBFormer [39] and PolyFormer [27]. The first six are mainstream GNNs and others are representative GTs. Implementation details are summarized in Appendix D.2.

### 5.3 Performance Comparison

To evaluate the model performance in node classification, we run each model ten times with random initializations. The results in terms of mean accuracy and standard deviation are reported in Table 1 and Table 2.

First, we can observe that SwapGT achieves the best performance on all datasets with different data splitting strategies, demonstrating the effectiveness of SwapGT in node classification. Then, we

Table 2: Comparison of all models in terms of mean accuracy $\pm$ stdev (%) under sparse splitting. The best results appear in **bold**. The second results appear in underline.

| Dataset | Photo | ACM | Computer | Citeseer | WikiCS | BlogCatalog | UAI2010 | Flickr |
|---|---|---|---|---|---|---|---|---|
| $\mathcal{H}$ | 0.83 | 0.82 | 0.78 | 0.74 | 0.66 | 0.40 | 0.36 | 0.24 |
| SGC | $91.90_{\pm0.35}$ | $89.57_{\pm0.28}$ | $86.79_{\pm0.19}$ | $66.41_{\pm0.59}$ | $74.99_{\pm0.19}$ | $71.23_{\pm0.06}$ | $51.61_{\pm0.41}$ | $39.43_{\pm0.50}$ |
| APPNP | $92.24_{\pm0.28}$ | $89.91_{\pm0.89}$ | $87.64_{\pm0.39}$ | $66.70_{\pm0.11}$ | $77.42_{\pm0.31}$ | $81.76_{\pm0.38}$ | $61.65_{\pm0.71}$ | $71.39_{\pm0.62}$ |
| GPRGNN | $92.13_{\pm0.32}$ | $89.47_{\pm0.90}$ | $86.38_{\pm0.44}$ | $66.50_{\pm0.62}$ | $77.59_{\pm0.49}$ | $84.57_{\pm0.35}$ | $58.75_{\pm0.75}$ | $71.89_{\pm0.89}$ |
| FAGCN | $92.02_{\pm0.18}$ | $88.47_{\pm0.31}$ | $83.99_{\pm1.95}$ | $64.54_{\pm0.66}$ | $75.21_{\pm0.84}$ | $76.38_{\pm0.82}$ | $54.67_{\pm0.96}$ | $63.68_{\pm0.72}$ |
| BM-GCN | $91.19_{\pm0.39}$ | $90.11_{\pm0.60}$ | $86.14_{\pm0.51}$ | $66.11_{\pm0.47}$ | $77.39_{\pm0.37}$ | $84.05_{\pm0.54}$ | $57.51_{\pm1.14}$ | $60.82_{\pm0.76}$ |
| ACM-GCN | $91.71_{\pm0.64}$ | $89.68_{\pm0.45}$ | $86.64_{\pm0.59}$ | $64.85_{\pm1.19}$ | $77.68_{\pm0.57}$ | $77.17_{\pm1.34}$ | $56.05_{\pm2.11}$ | $64.58_{\pm1.53}$ |
| ANS-GT | $90.42_{\pm0.74}$ | $88.58_{\pm0.68}$ | $84.95_{\pm0.37}$ | $62.94_{\pm1.16}$ | $75.72_{\pm0.47}$ | $78.26_{\pm0.53}$ | $55.85_{\pm1.64}$ | $64.42_{\pm1.28}$ |
| NAGphormer | $91.65_{\pm0.80}$ | $89.73_{\pm0.48}$ | $85.31_{\pm0.65}$ | $63.66_{\pm1.68}$ | $76.93_{\pm0.75}$ | $79.19_{\pm0.41}$ | $58.36_{\pm1.01}$ | $67.48_{\pm1.04}$ |
| SGFormer | $90.13_{\pm0.56}$ | $88.03_{\pm0.60}$ | $80.07_{\pm0.21}$ | $62.41_{\pm0.94}$ | $74.69_{\pm0.52}$ | $78.15_{\pm0.69}$ | $50.19_{\pm1.72}$ | $51.01_{\pm1.05}$ |
| Specformer | $90.57_{\pm0.55}$ | $88.20_{\pm1.05}$ | $85.55_{\pm0.63}$ | $62.64_{\pm1.54}$ | $75.24_{\pm0.71}$ | $79.75_{\pm1.29}$ | $57.42_{\pm1.06}$ | $56.94_{\pm1.48}$ |
| VCR-Graphormer | $91.39_{\pm0.75}$ | $86.81_{\pm0.84}$ | $85.06_{\pm0.64}$ | $57.61_{\pm0.60}$ | $72.81_{\pm1.44}$ | $74.90_{\pm1.18}$ | $56.43_{\pm1.10}$ | $50.93_{\pm1.12}$ |
| CoBFormer | $91.21_{\pm0.62}$ | $89.18_{\pm0.89}$ | $85.06_{\pm0.79}$ | $63.82_{\pm1.34}$ | $76.28_{\pm1.28}$ | $79.44_{\pm0.98}$ | $58.23_{\pm0.82}$ | $66.94_{\pm1.18}$ |
| PolyFormer | $91.52_{\pm0.78}$ | $89.83_{\pm0.62}$ | $85.75_{\pm0.78}$ | $64.77_{\pm1.27}$ | $75.12_{\pm1.16}$ | $81.02_{\pm0.81}$ | $58.89_{\pm0.77}$ | $67.85_{\pm1.43}$ |
| SwapGT | $\mathbf{92.93}_{\pm0.26}$ | $\mathbf{90.92}_{\pm0.69}$ | $\mathbf{88.14}_{\pm0.52}$ | $\mathbf{69.91}_{\pm1.02}$ | $\mathbf{78.11}_{\pm0.83}$ | $\mathbf{88.11}_{\pm0.58}$ | $\mathbf{63.96}_{\pm1.09}$ | $\mathbf{72.16}_{\pm1.19}$ |

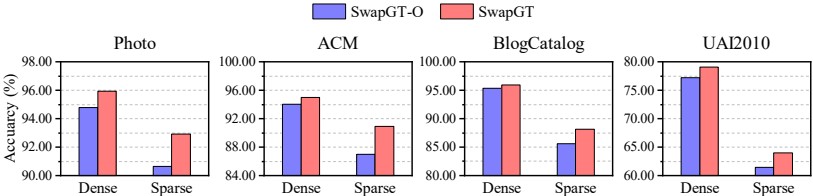

Figure 3: Performances of SwapGT with or without the center alignment loss.

can find that advanced GTs obtain more competitive performance than GNNs on over half datasets under dense splitting. But under sparse splitting, the situation reversed. An intuitive explanation is that Transformer has more learnable parameters than GNNs, which bring more powerful modeling capacity. However, it also requires more training data than GNNs in the training stage to ensure the performance. Therefore, when the training data is sufficient, GTs can achieve promising performance. And when the training data is sparse, GTs usually leg behind GNNs. Our proposed SwapGT addresses this issue by introducing the token swapping operation to generate diverse token sequences. This operation effectively augments the training data, ensuring the model training even in the sparse data scenario. In addition, the tailored center alignment loss also constrains the model parameter learning, further enhancing the model performance.

Moreover, the performance gain of SwapGT in the sparse setting is larger than in the dense setting. Specifically, SwapGT can achieve around 3.5% and 3.2% performance improvement on BlogCatalog and Citeseer, respectively. This is because the token swapping operation in SwapGT can leverage the semantic relevance of node tokens to generate more informative token sequences, which could be regarded as a data augmentation strategy. These augmented token sequences can improve the data utilization efficiency and significantly enhance the quality of node representation learning, especially in the data sparse splitting. Hence, SwapGT can bring more significant improvement in the sparse splitting.

### 5.4 Study on the center alignment loss

The center alignment loss, proposed to constrain the representation learning from multiple token sequences, is a key design of SwapGT. Here, we validate the effectiveness of the center alignment loss in node classification. Specifically, we develop a variant of SwapGT by removing the center alignment loss, called SwapGT-O. Then, we evaluate the performance of SwapGT-O on all datasets under dense splitting and sparse splitting. Due to the space limitation, we only report the results on four datasets in Figure 3, other results are reported in Appendix A.2. Based on the experimental results, we can have the following observations: 1) SwapGT beats SwapGT-O on all datasets, indicating that the developed center alignment loss can effectively enhance the performance of SwapGT. 2) Adopting the center alignment loss can bring more significant improvements in sparse setting than those in dense setting. This situation implies that introducing the reasonable constraint loss function based on the property of node token sequences can effectively improve the model training when the training data is sparse.

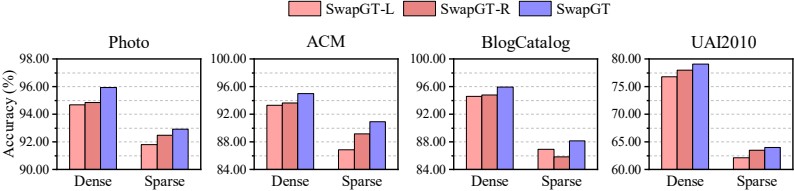

Figure 4: Performances of SwapGT with different token sequence generation strategies.

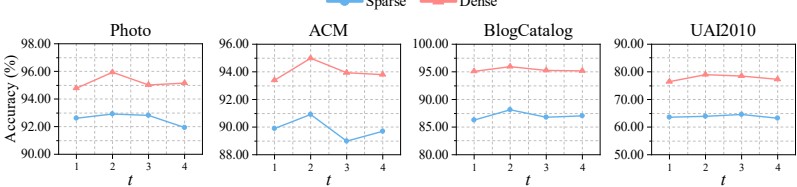

Figure 5: Analysis on the swapping times $t$.

## 5.5 Study on the token sequence generation

The generation of token sequences is another key module of SwapGT, which develops a novel token swapping operation can fully leverage the semantic relevance of nodes to generate informative token sequences. In this section, we evaluate the effectiveness of the proposed strategy by comparing it with two naive strategies. One is to enlarge the sampling size $k$. We propose a variant called SwapGT-L by sampling $2k$ tokens to construct token sequences. The other is to randomly sample $k$ tokens from the enlarged $2k$ token set to construct multiple token sequences, called SwapGT-R. Performance of these variants are shown in Figure 4 and results on other datasets are reported in Appendix A.3.

We can observe that SwapGT-R outperforms SwapGT-L on most cases, indicating that constructing multiple token sequences is better for node representation learning of tokenized GTs than generating single long token sequence. Moreover, SwapGT surpasses SwapGT-R on all cases, showcasing the superiority of the proposed token swapping operation in generation of multiple token sequences. This observation also implies that constructing informative token sequences can effectively improve the performance of tokenized GTs.

## 5.6 Analysis on the swapping times $t$

As discussed in Section 4.2, $t$ determines the range of candidate tokens from the constructed $k$-NN graph, further affecting the model performance. To validate the influence of $t$ on model performance, we vary $t$ in $\{1, 2, 3, 4\}$ and observe the changes of model performance. Results are shown in Figure 5 and Appendix A.5. We can clearly observe that SwapGT can achieve satisfied performance on all datasets when $t$ is no less than 2. This situation indicates that learning from tokens with semantic associations beyond the immediate neighbors can effectively enhancing the model performance. This phenomenon also reveals that reasonably enlarging the sampling space to seek more informative tokens is a promising way to improve the effect of node tokenized GTs.

## 5.7 Analysis on the augmentation times $s$

The augmentation times $s$ determines how many token sequences are adopted for node representation learning. Similar to $t$, we vary $s$ in $\{1, 2, \ldots, 8\}$ and report the performance of SwapGT. Results are shown in Figure 6 and Appendix A.6. Generally speaking, sparse splitting requires a larger $s$ to achieve the best performance, compared to dense splitting. This is because SwapGT needs more token sequences for model training in the sparse data scenario. This situation indicates that a tailored data augmentation strategy can effectively improve the performance of tokenized GTs when training data is sparse. Moreover, the optimal $s$ varies on different graphs. This is because different graphs exhibit different topology features and attribute features, which affects the generation of token sequences, further influencing the model performance.

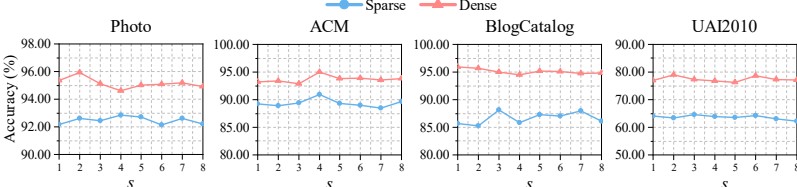

Figure 6: Analysis on the augmentation times $s$.

# 6 Conclusion

In this paper, we introduced a novel tokenized Graph Transformer SwapGT for node classification. In SwapGT, we developed a novel token swapping operation that flexibly swaps tokens in different token sequences, thereby generating diverse token sequences. This enhances the model's ability to capture rich node representations. Furthermore, SwapGT employs a tailored Transformer-based backbone with a center alignment loss to learn node representations from the generated multiple token sequences. The center alignment loss helps guide the learning process when nodes are associated with multiple token sequences, ensuring that the learned representations are consistent and informative. Experimental results demonstrate that SwapGT significantly improves node classification performance, outperforming several representative GT and GNN models.

## Acknowledgments

This work is supported by National Natural Science Foundation (U22B2017).

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

# A Additional Experimental Results

In this section, we provide the additional experimental results of ablation studies and parameter studies.

## A.1 Study of the propagation step

SwapGT leverages the graph diffusion strategy to calculate the topological features of nodes, which are further utilized to generate token sequences in the topology feature space. Here, we investigate the influence of the propagation step $K$ on the model performance. Specifically, we vary $K$ in $\{1, 2, 3, 4\}$ and observe the performance of SwapGT. The results under dense splitting and sparse splitting are shown in Figure 7 and Figure 8.

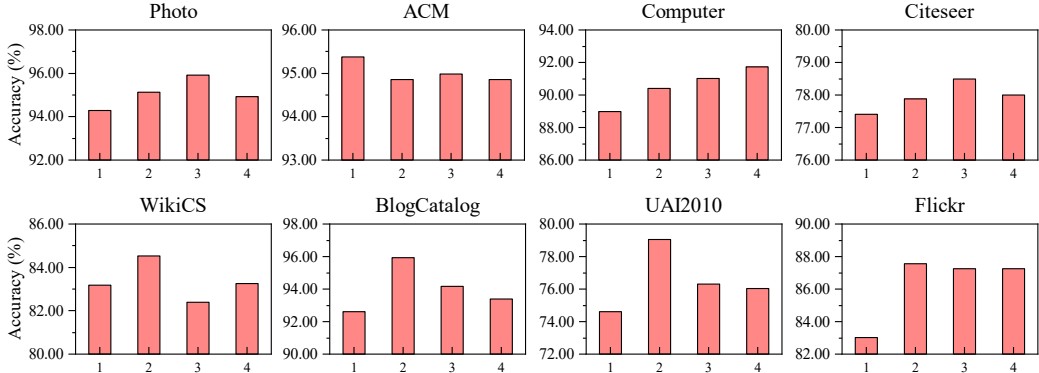

Figure 7: Performances of SwapGT with different propagation steps under dense splitting.

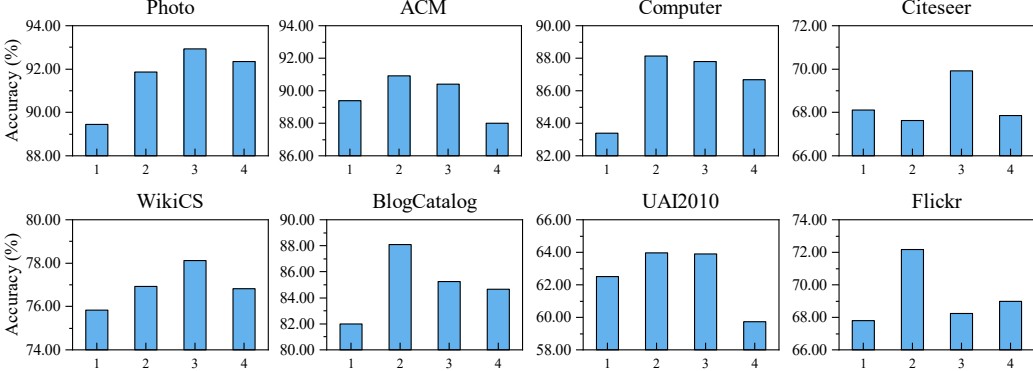

Figure 8: Performances of SwapGT with different propagation steps under sparse splitting.

Generally speaking, the influence of propagation steps is greater in the sparse splitting due to the heightened sensitivity of sparse splitting to input data during model training. Moreover, we can observe that the optimal propagation steps consistently fall within the range of 3 on almost all datasets under different splitting strategies. This is because that excessively large $K$ values risk over-smoothing while insufficient values fail to capture meaningful structural patterns. In practice, we utilize the grid search to determine the optimal $K$ within a constrained parameter space ($K<=4$).

## A.2 Study of the center alignment loss

The experimental results of SwapGT and SwapGT-O on the rest datasets are shown in Figure 9. We can observe that SwapGT outperforms SwapGT-O on most datasets. Moreover, the effect of applying the center alignment loss on SwapGT in sparse splitting is more significant than that in dense splitting. The above observations are in line with those reported in the main text. Therefore,

we can conclude that the center alignment loss can effectively enhance the performance of SwapGT in node classification.

## A.3    Study of the token sequence generation

The experimental results of SwapGT with different token sequence generation strategies on the rest datasets are shown in Figure 10. We can find that the additional experimental results exhibit similar observations shown in the main text. This situation demonstrates the effectiveness of the token sequence generation with the proposed token swapping operation in enhancing the performance of tokenized GTs. Moreover, we can also observe that the gains of introducing the token swapping operation vary on different graphs based on the results shown in Figure 4 and Figure 10. This phenomenon may attribute to that different graphs possess unique topology and attribute information, which further impact the selection of node tokens. While SwapGT applies the uniform strategy for selecting node tokens, which could lead to varying gains of token swapping. This situation also motivates us to consider different strategies of token selection on different graphs as the future work.

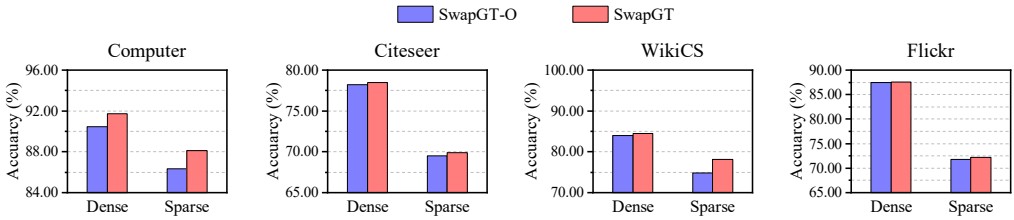

Figure 9:  Performances of SwapGT with or without the center alignment loss.

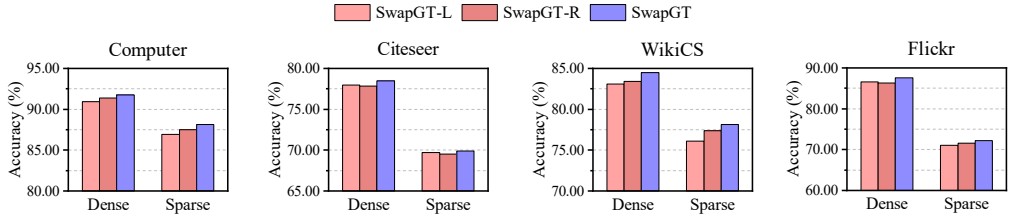

Figure 10:  Performances of SwapGT with different token sequence generation strategies.

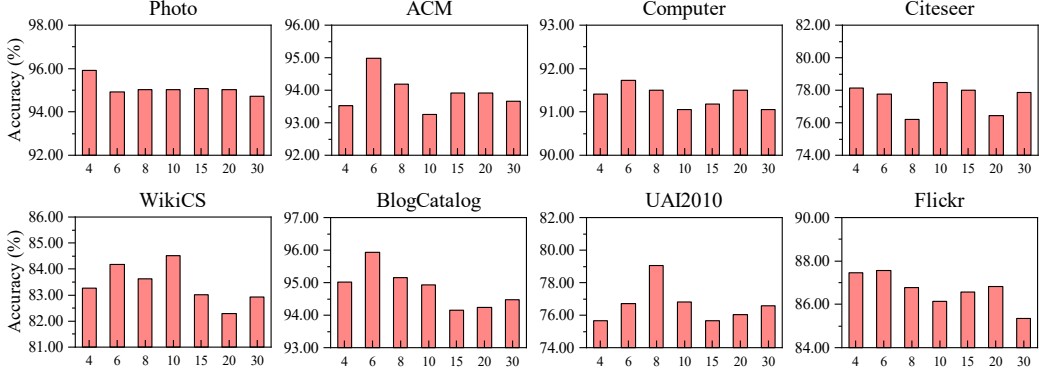

Figure 11:  Performances of SwapGT with different sampling sizes under dense splitting.

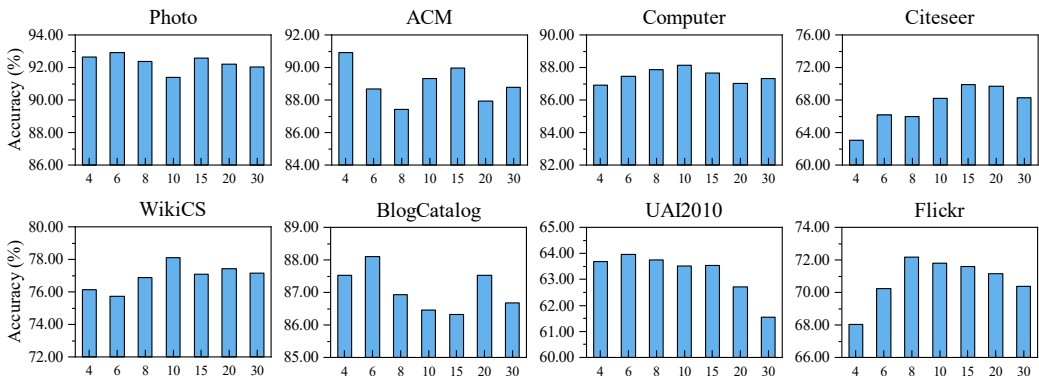

Figure 12: Performances of SwapGT with different sampling sizes under sparse splitting.

### A.4 Study of the token sampling size

To investigate the influence of token sampling size on model performance, we vary the sampling size in $\{4, 6, 8, 10, 15, 20, 30\}$ and report the corresponding results of SwapGT on all datasets under different splitting strategies. Figure 11 and Figure 12 show the results. We can observe that the best choice of the sampling size falls within 10 on almost all datasets, which implies that enlarging the length of token sequences cannot enhance the model performance on most cases whether in dense splitting or sparse splitting. This may because that a large sampling size is more easily to introduce irrelevant nodes as tokens, which further hurt the model performance.

### A.5 Analysis of the swapping times $t$

Here we report the rest results of SwapGT with varying $t$, which are shown in Figure 13. Similar to the phenomenons shown in Figure 5, SwapGT can achieve the best performance on all datasets when $t > 2$. Based on the results shown in Figure 13 and Figure 5, we can conclude that introducing tokens beyond first-order neighbors via the proposed token swapping operation can effective improve the performance of SwapGT in node classification.

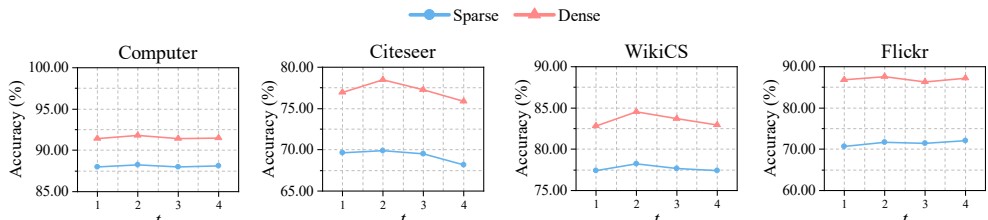

Figure 13: Performances of SwapGT with varying $t$.

### A.6 Analysis of the augmentation times $s$

Similar to analysis of $t$, the rest results of SwapGT with varying $s$ are shown in Figure 14. We can also observe the similar situations shown in Figure 6 that SwapGT requires a larger value of $s$ under sparse splitting compared to dense splitting. The situation demonstrates that introducing augmented token sequences can bring more significant performance gain in sparse splitting than that in dense splitting.

### A.7 Analysis of readout functions

The readout function is also an important module in SwapGT. According to Equation 7, we regard the combination of the raw token sequence and the augmented token sequences as the final node

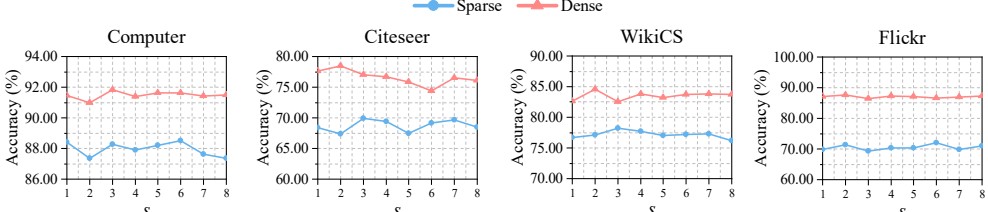

Figure 14: Performances of SwapGT with varying $s$.

Table 3: Comparison of SwapGT and its variant under dense splitting.

| Dataset | Photo | ACM | Computer | Citeseer | WikiCS | BlogCatalog | UAI2010 | Flickr |
|---|---|---|---|---|---|---|---|---|
| SwapGT-AT | 95.58 | 94.61 | 91.35 | 77.92 | 83.95 | 95.47 | 78.32 | 87.11 |
| SwapGT | 95.92 | 94.98 | 91.73 | 78.49 | 84.52 | 95.93 | 79.06 | 87.56 |

representation. This strategy can ensure the independence of the information from the original token sequence and generated token sequences. Moreover, we utilize the mean function to obtain the information of generated token sequences, which is a simple but efficient strategy and has been widely adopted in GNNs.

Here, we conduct experiments to validate the effectiveness of the proposed readout function, Specifically, we develop a variant named SwapGT-AT which leverages the attention-based readout function from NAGphormer[6] for node representation learning. The results are reported in Table 3 and Table 4.

We can observe that SwapGT beats this variant on all datasets. The reason may be that the neighborhood tokens in NAGphormer have varying importance. In this situation, the attention-based readout function can enhance the model performance. However, the augmented token sequences generated by the token swapping operation do not have this characteristic. Hence, the simple readout function may be more suitable for SwapGT.

## B  Illustration of Token Swapping

Here, we provide a toy example to understand the operation of token swapping. As shown in Figure 15, node 1 is the target node. We first select node 3 and consider the tokens in its token sequences as candidates. Then we select node 6 from the candidates to swap node 3, and construct the new token sequence.

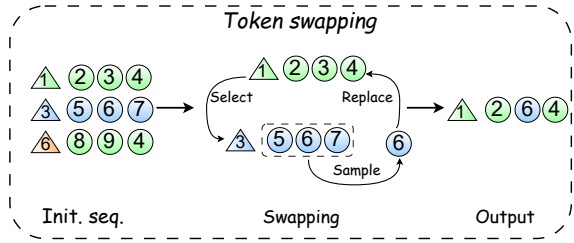

Figure 15: Illustration of the token swapping.

## C  Proof of Fact 1

Here we provide the detailed proof for **Fact 1**.

**Fact 1.** *If we guarantee that $(t + 1)p < 1$, in the generated token sequences, the probability of sampling lower-order neighbors is greater than sampling higher-order neighbors on the constructed*

Table 4: Comparison of SwapGT and its variant under sparse splitting.

| Dataset | Photo | ACM | Computer | Citeseer | WikiCS | BlogCatalog | UAI2010 | Flickr |
|---------|-------|------|----------|----------|--------|-------------|---------|--------|
| SwapGT-AT | 92.32 | 89.65 | 87.30 | 68.13 | 77.86 | 87.25 | 63.18 | 71.21 |
| SwapGT | 92.93 | 90.92 | 88.14 | 69.91 | 78.11 | 88.11 | 63.96 | 72.16 |

*k-NN graph, where $t$ denotes the maximum swapping times, and $p$ denotes the probability of swapping tokens per time in Algorithm 1.*

*Proof.* First, as discussed in Section 4.2, performing the swapping operation $t$ times enlarges the sampling space from 1-hop neighborhood to $(t+1)$-hop neighborhood on the constructed $k$-NN graph. Then, for an arbitrary node $i$, the probability of sampling its $m$-hop neighbors on the constructed $k$-NN graph (*i.e.*, swapping for $(m-1)$ times and maintaining for $(t-m+1)$ times) is described as follows:

$$P_i^m = C(t, m-1) \cdot p^{m-1} \cdot (1-p)^{t-m+1}. \tag{13}$$

Here, $C$ is the symbol of the combination number. In the augmented token sequences, neighbors of node $i$ from different $k$-NN hops all may be sampled. To reduce the noise caused by sampling too many high-hop neighbors on the constructed $k$-NN graph, the probability of sampling lower-order neighbors is expected to be greater, *i.e.*, $P_i^m > P_i^{m+1}$.

According to Equation 13,

$$P_i^m > P_i^{m+1} \iff C(t, m-1) \cdot p^{m-1} \cdot (1-p)^{t-m+1} > C(t, m) \cdot p^m \cdot (1-p)^{t-m}$$

$$\iff \frac{t!}{(m-1)!\,(t-m+1)!} \cdot p^{m-1} \cdot (1-p)^{t-m+1} > \frac{t!}{m!\,(t-m)!} \cdot p^m \cdot (1-p)^{t-m}$$

$$\iff \frac{1-p}{t-m+1} > \frac{p}{m}$$

$$\iff m > (t+1)\,p$$

where $1 \leq m, m+1 \leq t+1$, *i.e.*, $1 \leq m \leq t$. If we guarantee that $(t+1)p < 1$, it is obvious that $(t+1)p < 1 \leq m$. Finally, we can obtain **Fact 1**. $\qquad\square$

**Fact 1** suggests that with suitable maximum swapping times $t$ and probability of swapping tokens per time $p$, SwapGT is capable of well handling the potential noise and effectively enlarging the sampling space.

## D Experimental Settings

### D.1 Dataset

Here we introduce datasets adopted for experiments. The detailed statistics of all datasets are reported in Table 5.

- **Academic graphs**: This type of graph is formed by academic papers or authors and the citation relationships among them. Nodes in the graph represent academic papers or authors, and edges represent the citation relationships between papers or co-author relationships between two authors. The features of nodes are composed of bag-of-words vectors, which are extracted and generated from the abstracts and introductions of the academic papers. The labels of nodes correspond to the research fields of the academic papers or authors. ACM, Citeseer, WikiCS and UAI2010 belong to this type.

- **Co-purchase graphs**: This type of graph is constructed based on users' shopping behaviors. Nodes in the graph represent products. The edges between nodes indicate that two products are often purchased together. The features of nodes are composed of bag-of-words vectors extracted from product reviews. The category of a node corresponds to the type of goods the product belongs to. Computer and Photo belong to this type.

- **Social graphs**: This type of graph is formed by the activity records of users on social platforms. Nodes in the graph represent users on the social platform. The edges between nodes indicate the social relations between two users. Node features represent the text information extracted from the authors' homepage. The label of a node refers to the interest groups of users. BlogCatalog and Flickr belong to this type.

Table 5: Statistics of datasets, ranked by the homophily level.

| Dataset | # nodes | # edges | # features | # labels | $\mathcal{H} \downarrow$ |
|---------|---------|---------|------------|----------|------|
| Photo | 7,650 | 238,163 | 745 | 8 | 0.83 |
| ACM | 3,025 | 1,3128 | 1,870 | 3 | 0.82 |
| Computer | 13,752 | 491,722 | 767 | 10 | 0.78 |
| Citeseer | 3,327 | 4,552 | 3,703 | 6 | 0.74 |
| WikiCS | 11,701 | 216,123 | 300 | 10 | 0.66 |
| BlogCatalog | 5,196 | 171,743 | 8,189 | 6 | 0.40 |
| UAI2010 | 3,067 | 28,311 | 4,973 | 19 | 0.36 |
| Flickr | 7,575 | 239,738 | 12,047 | 9 | 0.24 |

## D.2 Implementation Details

For baselines, we refer to their official implementations and conduct a systematic tuning process on each dataset. For SwapGT, we employ a grid search strategy to identify the optimal parameter settings. Specifically, We try the learning rate in $\{0.001, 0.005, 0.01\}$, dropout in $\{0.3, 0.5, 0.7\}$, dimension of hidden representations in $\{256, 512\}$, $k$ in $\{4, 6, 8\}$, $\alpha$ in $\{0.1, \ldots, 0.9\}$. All experiments are implemented using Python 3.8, PyTorch 1.11, and CUDA 11.0 and executed on a Linux server with an Intel Xeon Silver 4210 processor, 256 GB of RAM, and a 2080TI GPU.

## D.3 Baselines

In this paper, we select mainstream GNNs and GTs as baselines. For GNNs, there are two categories of them, coupled GNNs and decoupled GNNs. FAGCN [3], BM-GCN [18], ACM-GCN [26] are recent coupled GNNs for node classification. SGC [36], APPNP [24], GPRGNN [15] are representative decoupled GNNs. For GTs, we select methods from tokenized GTs and hybrid GTs. ANS-GT [44], NAGphormer [6], VCR-Graphormer [17] and PolyFormer [27] are powerful tokenized GTs. While SGFormer [38], Specformer [2] and CoBFormer [39] are representative hybrid GTs.

# E   Limitation of SwapGT

A potential limitation in SwapGT could be that SwapGT applies a uniform swapping probability $p$ to all tokens in the sequence. A hierarchical probability swapping framework may be a better solution, where nodes within the top-$k$ most relevant subset (e.g., the top $k/3$ nodes) are assigned higher swapping probabilities, while others receive lower probabilities. This refinement could mitigate noise interference.

