# OpenReview forum: "Rethinking Tokenized Graph Transformers  for Node Classification"
_NeurIPS.cc/2025/Conference — NeurIPS 2025 poster_

### Official Review · Reviewer_chKg · 2025-06-30

**Clarity:** 4
**Significance:** 3
**Originality:** 4
**Rating:** 5
**Confidence:** 5

**Summary:**

The tokenized graph Transformer has been a popular architecture of graph Transformers for node classification in recent years. While this paper reveals a common limitation of existing tokenized graph Transformer that these methods only leverage limited graph information to construct the token sequences, resulting in unsatisfied performance in several situations (e.g., sparse training data). To address this issue, the authors propose a new method called SwapGT which introduces a novel operation, token swapping. This proposed operation can generate diverse token sequences based on the initial token sequence, like a data augmentation strategy. In the end, the generated token sequences enhance the performance of graph Transformers for node classification, especially in the data sparsity situation.

**Questions:**

1.	Based on the previous tokenized graph Transformers (e.g., NAGphormer), the readout function is also an important module. However, the proposed readout function in Eq. (7) lacks necessary motivations. Please provide the insights for the proposed strategy.
2.	Moreover, do authors try other readout functions, such as attention-based method in NAGphormer?
3.	Although the experimental results in both dense splitting and sparse splitting show the promising results, these results lack deeper discussion. For instance, I notice that SwapGT show more significant improvement in the sparse splitting. Can you explain?
4.	There is no introduction for baselines. Why do you choose these methods? Can you provide the corresponding reasons?

**Ethical Concerns:**

["NO or VERY MINOR ethics concerns only"]

**Final Justification:**

In the rebuttal, the authors address my main concern about the motivation of the key modules in their proposed method. I'd like to maintain the score as 5.

**Limitations:**

Yes.

**Quality:**

3

**Strengths And Weaknesses:**

Strengths:

1.	This paper is well organized and easy to follow.
2.	The proposed token swapping operation is novel.
3.	The authors provide the corresponding theoretical analysis of the generated token sequences.
4.	The empirical results show promising performance.

Weaknesses:

1.	Some modules lack motivation.
2.	Experimental results need deeper and more insightful discussions.
3.	The baselines need necessary introductions, and the reason of selecting baselines is also required.

---

> ### Author Rebuttal · Authors · 2025-07-31
>
> We appreciate the reviewer for providing positive feedback and valuable comments on our paper. Following are our detailed responses to your questions.
>
> **Q1**. Based on the previous tokenized graph Transformers (e.g., NAGphormer), the readout function is also an important module. However, the proposed readout function in Eq. (7) lacks necessary motivations. Please provide the insights for the proposed strategy.
>
> **A1**. Thank you for your insightful suggestions. According to Eq. (7), we regard the combination of the raw token sequence and the augmented token sequences as the final node representation. This strategy can ensure the independence of the information from the original token sequence and generated token sequences. Moreover, we utilize the mean function to obtain the information of generated token sequences, which is a simple but efficient strategy and has been widely adopted in GNNs, such as GraphSAGE.
>
> We will add the above discussions in the revised version to highlight the motivation of the readout function in SwapGT.
>
> **Q2**. Moreover, do authors try other readout functions, such as attention-based method in NAGphormer?
>
> **A2**. Thank you for your helpful questions. Per your suggestion, we develop a variant named SwapGT-AT which leverages the attention-based readout function for node representation learning. The results are as follows:
>
> | Sparse    | Photo | ACM   | Computer | Citeseer | Blogcatalog | UAI   | Flickr | wikics |
> |-----------|-------|-------|----------|----------|-------------|-------|--------|--------|
> | SwapGT-AT | 92.32 | 89.65 | 87.30    | 68.13    | 87.25       | 63.18 | 71.21  | 77.86  |
> | SwapGT    | 92.93 | 90.92 | 88.14    | 69.91    | 88.11       | 63.96 | 72.16  | 78.11  |
>
> | Dense     | Photo | ACM   | Computer | Citeseer | Blogcatalog | UAI   | Flickr | wikics |
> |-----------|-------|-------|----------|----------|-------------|-------|--------|--------|
> | SwapGT-AT | 95.58 | 94.61 | 91.35    | 77.92    | 95.47       | 78.32 | 87.11  | 83.95  |
> | SwapGT    | 95.92 | 94.98 | 91.73    | 78.49    | 95.93       | 79.06 | 87.56  | 84.52  |
>
> We can observe that SwapGT beats this variant on all datasets. The reason may be that the neighborhood tokens in NAGphormer have varying importance. In this situation, the attention-based readout function can enhance the model performance. However, the augmented token sequences generated by the token swapping operation do not have this characteristic. Hence, the simple readout function may be more suitable for SwapGT.
>
> We will add the above results as well as the discussions in the revised version.
>
>
> **Q3**. Although the experimental results in both dense splitting and sparse splitting show the promising results, these results lack deeper discussion. For instance, I notice that SwapGT show more significant improvement in the sparse splitting. Can you explain?
>
> **A3**. Thank you for your insightful questions. As discussed in Line 241-249, the token swapping operation in SwapGT can leverage the semantic relevance of node tokens to generate more informative token sequences, which could be regarded as a data augmentation strategy. These augmented token sequences can improve the data utilization efficiency and significantly enhance the quality of node representation learning, especially in the data sparse splitting. Hence, SwapGT can bring more significant improvement in the sparse splitting.
>
> We will update the discussions about the results of performance comparison based on the above content in the revised version.
>
>
> **Q4**. There is no introduction for baselines. Why do you choose these methods? Can you provide the corresponding reasons?
>
>
> **A4**. Thank you for your helpful questions. In this paper, we select mainstream GNNs and GTs as baselines. For GNNs, there are two categories of them, coupled GNNs and decoupled GNNs. FAGCN, BM-GCN, ACM-GCN are recent coupled GNNs for node classification. SGC, APPNP, GPRGNN are representative decoupled GNNs. For GTs, we select methods from tokenized GTs and hybrid GTs. NAGphormer, VCR-Graphormer and PolyFormer are powerful tokenized GTs. While SGFormer, Specformer and CoBFormer are representative hybrid GTs.
>
> We will add the above discussions about the baselines in the revised version.

---

> > ### Comment · Reviewer_chKg · 2025-08-02
> >
> > Thank you for your detailed responses, which address my concerns. I will maintain the positive score and hope the authors can include these comparisons and discussions in your revised paper.

---

> > > ### Author Response · Authors · 2025-08-03
> > >
> > > Many thanks for your positive feedback that greatly encourages us. We will carefully revise the paper according to the reviewers' suggestions and additional experimental results.

---

### Official Review · Reviewer_ZA1z · 2025-07-01

**Clarity:** 3
**Significance:** 2
**Originality:** 3
**Rating:** 4
**Confidence:** 4

**Summary:**

This paper proposes a novel tokenized Graph Transformer for node classification by developing a novel token swapping operation that flexibly swaps tokens in different token sequences,called SwapGT,significantly outperforming several GTs and GNNs on extensive experiments.

**Questions:**

See the weakness part.

**Ethical Concerns:**

["NO or VERY MINOR ethics concerns only"]

**Final Justification:**

The authors' response has addressed my concerns on novelty and experiments. Therefore, I decide to raise my score.

**Limitations:**

No. The authors do not adequately address the limitations of their study and its potential future optimization directions of their work.

- Cons of the novel token swapping operation proposed by SwapGT?

- Any other more efficient method to enhances the model’s ability to capture rich node representations?

**Paper Formatting Concerns:**

No.

**Quality:**

2

**Strengths And Weaknesses:**

Strengths:
- S1. The method proposed in this paper enhances the model’s ability to capture rich node representations.
- S2. Experiments on real-world datasets demonstrate that SwapGT achieves better performance on node classification, compared with several representative GNNs and GTs.

Weaknesses:
- W1. Limited Novelty. The novelty is not clearly articulated, and the distinction from prior tokenized GTs needs further clarification. This paper also lacks a discussion between the proposed method and GTs with global attention.
- W2. Performance Improvement. Although the proposed SwapGT addresses the performance degradation in sparse splitting, it seems that GNNs do not suffer from sparse splitting, which weakens the significance of performance improvement.
- W3. Missing limitations. The limitations of the proposed method are not adequately discussed, potentially limiting readers’ understanding of its scope and applicability.

---

> ### Author Rebuttal · Authors · 2025-07-31
>
> Thank you for detailed comments. We provide the following responses:
>
> **Q1**. Limited Novelty. The novelty is not clearly articulated, and the distinction from prior tokenized GTs needs further clarification. This paper also lacks a discussion between the proposed method and GTs with global attention.
>
> **A1**. Thank you for your comments. First, regarding the differences between SwapGT and prior node-tokenized GTs: as outlined in Lines 45–61 and 109–115, existing node tokenized GTs typically construct token sequences by statically selecting 1-hop neighbors of target nodes from the k-NN graph. In contrast, SwapGT introduces a critical innovation through its token swapping operation, which dynamically expands token sequences beyond the fixed 1-hop neighborhood. By incorporating nodes from broader regions of the k-NN graph (i.e., beyond direct 1-hop connections) into token sequences, SwapGT generates more diverse and contextually rich token sets. This flexibility allows the model to capture more informative structural patterns, which we demonstrate empirically enhances performance in node classification. Besides, the novelty of the proposed SwapGT has also been recognized by all other reviewers.
>
> Then, as discussed in Line 27-35 and Line 94-108, GTs with global attention usually adopts the design of combining Transformer modules with GNN-style modules to construct hybrid neural network layers, which requires the entire graph as the model input and calculating the attention scores between all nodes, leading to the over-globalization issue. While SwapGT adopts the tokenized design, which transforms the input graph into independent token sequences for node representation learning. Since tokenized GTs only focus on the generated tokens, they naturally avoiding the over-globalization issue.
>
> In the revised version, we will update the discussions of SwapGT and prior GTs, including tokenized GTs and GTs with global attention to highlight the novelty of SwapGT.
>
> **Q2**. Performance Improvement. Although the proposed SwapGT addresses the performance degradation in sparse splitting, it seems that GNNs do not suffer from sparse splitting, which weakens the significance of performance improvement.
>
> **A2**. Thank you for your comments. According to the results shown in Table 2, SwapGT gains 0.5% performance improvement over the runner-up baseline on most datasets. Specifically, SwapGT can achieve around 3.5% and 3.2% performance improvement on BlogCatalog and Citeseer, respectively. These results showcase that the designs of SwapGT can bring significant performance improvement for the node classification task.
>
> We will add the above discussions in the revised version to highlight the significance of performance improvement.
>
> **Q3**. Missing limitations. The limitations of the proposed method are not adequately discussed, potentially limiting readers’ understanding of its scope and applicability.
>
> **A3**. Thank you for your comments. In the submitted version, we have discussed the limitation of SwapGT in the Appendix E. The potential limitation in SwapGT could be that SwapGT applies a uniform swapping probability $p$ to all tokens in the sequence which ignores the importance of different node tokens. A hierarchical probability swapping framework may be a better solution, where nodes within the top-k most relevant subset are assigned higher swapping probabilities, while others receive lower probabilities. This refinement could mitigate noise interference. In addition, your two suggestions are also valuable for discussing the limitation of SwapGT.
>
> Based on the above content and your comments, we will update the discussions about the limitations of SwapGT in the revised version.

---

> > ### Comment · Reviewer_ZA1z · 2025-08-03
> >
> > I appreciate the authors' detailed response. However, the answers provided closely resemble the content presented in the paper and do not sufficiently address my concerns regarding the novelty and distinguishability of the proposed method. Consequently, I have decided to maintain my score.

---

> > > ### Author Response · Authors · 2025-08-04
> > >
> > > Thank you for your response. We have reorganized and summarized the distinctions between SwapGT and prior methods, including graph transformers (GTs) with global attention and tokenized GTs, as follows:
> > >
> > > - The core distinction between SwapGT and GTs with global attention lies in their input modalities: SwapGT takes sampled token sequences as input, such that it only computes attention scores between tokens within the input sequence. In contrast, GTs with global attention take the entire graph as input, requiring them to compute attention scores across all nodes. Computing attention scores within tokens in the input token sequence can naturally overcome the over-globalization issue in GTs with global attention.
> > >
> > > - The key distinction between SwapGT and prior tokenized GTs resides in their token generators. Prior methods employ a top-$k$ sampling strategy, which selects only a small subset of nodes to construct token sequences. SwapGT, by contrast, utilizes our proposed token swapping operation, which leverages semantic relevance between node tokens to include more node tokens with potential semantic relevance to generate diverse token sequences.
> > >
> > > - The core contribution of SwapGT is the development of a novel token swapping operation, which fully exploits relational information between node tokens to capture potentially relevant nodes beyond top-$k$ sampling and generate new token sequences, surpassing the capabilities of prior token generators.
> > >
> > > We have endeavored to clarify the distinctions between SwapGT and prior methods, as well as the novelty of our approach. If the above still fails to address your concerns regarding the novelty and distinctiveness of the proposed method, could you specify in detail the specific issues within the content that prevent it from addressing your concerns regarding innovation and distinguishability, as well as the specific details of your concerns in this regard? This would enable a more detailed and thorough discussion.
> > >
> > > Thank you again for your attention.

---

> > > > ### Comment · Reviewer_ZA1z · 2025-08-05
> > > >
> > > > Thank you for your detailed explanation. I understand the differences between the proposed **token swap** strategy and other **token fixing** strategies. However, the proposed strategy appears to be a lightweight version of graph transformers (GTs) with global attention, which attends to $k$-hop neighbors with controllable probabilities. It lacks a substantial theoretical improvement compared to GTs that utilize $1$-hop neighbors and those with global attention. Therefore, I believe a borderline rejec is an appropriate assessment.

---

> > > > > ### Author Response · Authors · 2025-08-05
> > > > >
> > > > > First, we would like to clarify that in the node classification task, tokenized GTs [1,2,3,4] and GTs with global attention [5,6,7,8] **are two distinct categories of methods**. GTs with global attention require designing graph structure biases or incorporating GNN-like models because Transformer cannot preserve graph structural information via the original global attention. In contrast, tokenized GTs only need to design different token generators and can effectively learn node representations from input token sequences using standard Transformers. Thus, these are two separate types of methods.
> > > > >
> > > > > Then, this paper first analyzes that the token generation strategies of existing node tokenized GTs are equivalent to selecting 1-hop nodes on the constructed K-NN graph to build token sequences. This strategy can only retain limited graph information, which inevitably hinders the model's learning of node representations, especially in sparse scenarios (as verified by our experimental results). Moreover, GTs with global attention are also affected by data sparsity, and their performance is generally poor.
> > > > >
> > > > > To address the limitation of GTs in node classification, we propose SwapGT, which introduces a novel token swapping operation to generate diverse token sequences. The designs of SwapGT can significantly improve the model's performance in node classification, particularly in sparse scenarios.
> > > > >
> > > > > Based on the above discussions, our method is significantly different from existing approaches (tokenized GTs and GTs with global attention), and can effectively address the limitations of GTs for node classification in sparse scenarios.
> > > > >
> > > > >
> > > > > [1] NAGphormer: A tokenized graph transformer for node classification in large graphs. ICLR 2023.
> > > > >
> > > > > [2] Polyformer: Scalable node-wise filters via polynomial graph transformer. KDD 2024.
> > > > >
> > > > > [3] VCR-Graphormer: A mini-batch graph transformer via virtual connections. ICLR 2024.
> > > > >
> > > > > [4] Leveraging Contrastive Learning for Enhanced Node Representations in Tokenized Graph Transformers. NeurIPS 2024.
> > > > >
> > > > > [5] Nodeformer: A scalable graph structure learning transformer for node classification. NeurIPS 2022.
> > > > >
> > > > > [6] SGFormer: Simplifying and empowering transformers for large-graph representations. NeurIPS 2023.
> > > > >
> > > > > [7] Specformer: Spectral graph neural networks meet transformers. ICLR 2023.
> > > > >
> > > > > [8] DUALFormer: Dual Graph Transformer. ICLR 2025.

---

> > > > > > ### Comment · Reviewer_ZA1z · 2025-08-05
> > > > > >
> > > > > > Thanks for your clarification. I have no problem on the novelty now. I have correspondingly raised my score.

---

> > > > > > > ### Author Response · Authors · 2025-08-05
> > > > > > >
> > > > > > > Thanks for your patient discussions and positive feedbacks. We will carefully prepare the revised version based on the above discussions. Thank you for your time!

---

### Official Review · Reviewer_n87V · 2025-07-02

**Clarity:** 3
**Significance:** 3
**Originality:** 4
**Rating:** 5
**Confidence:** 5

**Summary:**

This paper proposes a novel tokenized graph Transformer named SwapGT for the node classification task. Specifically, the authors firstly analyze the drawbacks of token generation in existing tokenized graph Transformers. Then, they design a new token generator called token swapping. By swapping tokens between different token sequences, the proposed token generator can generate diverse token sequences which carry rich graph information. In addition, the authors also develop a center-alignment loss to help the model learn node representations from multi-token sequences. Extensive results indicate that SwapGT outperforms other mainstream methods in the node classification task, such as GNNs and GTs.

**Questions:**

The questions are summarized as follows:

1. According to Fig. 2, the proposed SwapGT can be regarded as a generalized framework where the Transformer layer is just utilized to learn node representations from input token sequences. In this way, the learning module is optional. I am curious about the performance of SwapGT when integrated with other Transformer variants, such as linear Transformer and PolyFormer.
2. Based on Q1, I suggest authors conduct additional ablation studies to validate the influence of learning module on model performance, just like the ablation studies in GraphGPS.
3. I have noticed that $\alpha$ in Eq. (8) is an important hyper-parameter. So, can you provide the necessary experiments to explore the influence of $\alpha$ on model performance?
4. In A.4, the authors provide the results of SwapGT with different token sampling size. I have found that sparse splitting prefers larger sampling size. Can you provide some insightful discussions for this phenomenon?
5. What is the value of the swapping probability $p$ in practice?

**Ethical Concerns:**

["NO or VERY MINOR ethics concerns only"]

**Final Justification:**

The authors provide the additional empirical results of ablation study and parameter analysis which address my concerns raised in the rebuttal. And I also review the comments of other reviewers. Finally, I decide to keep my score.

**Limitations:**

Yes.

**Paper Formatting Concerns:**

There is no paper formatting concern in this paper.

**Quality:**

3

**Strengths And Weaknesses:**

The strengths and weaknesses are summarized as follows:

Strengths:
1. The motivation of SwapGT is clear.
2. The idea of swapping token between different token sequences is new and interesting.
3. The paper is well written.
4. The experiments of performance comparison are extensive.

Weaknesses:
1. The authors only consider standard Transformer.
2. The experimental section lacks additional ablation studies.
3. The analysis of the key hyper-parameter is also required.

---

> ### Author Rebuttal · Authors · 2025-07-31
>
> We appreciate the reviewer for providing positive feedback and valuable comments on our paper. Following are our detailed responses to your questions.
>
> **Q1&Q2**. According to Fig. 2, the proposed SwapGT can be regarded as a generalized framework where the Transformer layer is just utilized to learn node representations from input token sequences. In this way, the learning module is optional. I am curious about the performance of SwapGT when integrated with other Transformer variants, such as linear Transformer and PolyFormer? Based on Q1, I suggest authors conduct additional ablation studies to validate the influence of learning module on model performance, just like the ablation studies in GraphGPS.
>
> **A1&A2**. Thank you for your insightful questions. Per your suggestions, we develop two variants of SwapGT, named SwapGT-LT and SwapGT-PF. The former replace the standard Transformer module with the linear attention-based version. The latter introduce the signed attention-based learning module introduced by PolyFormer for representation learning. The results are as follows:
>
> |   Sparse  | Photo |  ACM  | Computer | Citeseer | Blogcatalog |  UAI  | Flickr | wikics |
> |:---------:|:-----:|:-----:|:--------:|:--------:|:-----------:|:-----:|:------:|:------:|
> | SwapGT-LT | 91.35 | 89.06 |   85.69  |   67.21  |    83.73    | 58.22 |  65.13 |  75.13 |
> | SwapGT-PF | 92.08 | 90.15 |   87.03  |   68.74  |    86.97    | 62.92 |  71.44 |  77.32 |
> |   SwapGT  | 92.93 | 90.92 |   88.14  |   69.91  |    88.11    | 63.96 |  72.16 |  78.11 |
>
> |   Dense   | Photo |  ACM  | Computer | Citeseer | Blogcatalog |  UAI  | Flickr | wikics |
> |:---------:|:-----:|:-----:|:--------:|:--------:|:-----------:|:-----:|:------:|:------:|
> | SwapGT-LT | 94.16 | 94.07 |   86.26  |   77.98  |    94.41    | 74.03 |  82.44 |  81.23 |
> | SwapGT-PF | 95.65 | 94.42 |   91.09  |   78.17  |    95.37    | 78.25 |  87.24 |  84.08 |
> |   SwapGT  | 95.92 | 94.98 |   91.73  |   78.49  |    95.93    | 79.06 |  87.56 |  84.52 |
>
> We can observe that SwapGT beats these two variants on all datasets. This situation indicates that the standard Transformer module is capable of learning informative node representations in the tokenized graph Transformer architecture.
>
> We will add the above results and discussions in the revised version. Thank you.
>
> **Q3**. I have noticed that $\alpha$ in Eq. (8) is an important hyper-parameter. So, can you provide the necessary experiments to explore the influence of $\alpha$ on model performance?
>
> **A3**. Thank you for your insightful questions. Following your suggestions, we vary $\alpha$ in {0, 0.1, …, 1} and observe the performance of SwapGT. The results are as follows:
>
> | sparse | Photo  | ACM    | Computer | Citeseer | WikiCS | Blogcatalog | UAI2010 | Flickr |
> |--------|--------|--------|----------|----------|--------|-------------|---------|--------|
> | 0      | 92.87  | 88.72  | 87.75    | 68.56    | 76.71  | 86.25       | 59.63   | 71.98  |
> | 0.1    | 92.93  | 90.92  | 88.14    | 68.74    | 76.39  | 88.11       | 60.45   | 72.12  |
> | 0.2    | 92.55  | 89.18  | 87.68    | 69.59    | 78.11  | 84.97       | 61.31   | 72.13  |
> | 0.3    | 92.03  | 88.86  | 86.83    | 69.78    | 76.75  | 86.15       | 61.45   | 72.16  |
> | 0.4    | 91.32  | 87.64  | 85.62    | 69.78    | 76.26  | 85.76       | 63.82   | 72.13  |
> | 0.5    | 89.95  | 86.88  | 85.20    | 68.86    | 75.61  | 85.16       | 63.96   | 72.05  |
> | 0.6    | 86.93  | 86.91  | 82.94    | 69.62    | 73.96  | 85.76       | 63.68   | 71.98  |
> | 0.7    | 86.86  | 85.17  | 82.17    | 69.46    | 73.97  | 85.34       | 57.88   | 71.92  |
> | 0.8    | 83.75  | 83.26  | 80.11    | 69.75    | 72.38  | 84.59       | 56.68   | 71.86  |
> | 0.9    | 81.59  | 81.73  | 77.47    | 69.91    | 70.43  | 83.84       | 54.61   | 70.66  |
> | 1      | 78.92  | 78.42  | 74.05    | 69.81    | 70.07  | 82.77       | 54.37   | 71.62  |
>
> | dense | Photo  | ACM    | Computer | Citeseer | WikiCS | Blogcatalog | UAI2010 | Flickr |
> |-------|--------|--------|----------|----------|--------|-------------|---------|--------|
> | 0     | 94.98  | 93.66  | 91.44    | 78.37    | 84.49  | 94.77       | 77.36   | 86.03  |
> | 0.1   | 95.92  | 93.79  | 91.73    | 78.49    | 84.52  | 94.16       | 77.62   | 86.61  |
> | 0.2   | 94.56  | 94.59  | 91.15    | 78.37    | 84.04  | 94.08       | 79.06   | 86.77  |
> | 0.3   | 95.19  | 94.85  | 91.06    | 78.37    | 83.98  | 94.39       | 78.14   | 86.66  |
> | 0.4   | 94.98  | 94.85  | 90.48    | 78.37    | 83.87  | 95.93       | 77.36   | 87.40  |
> | 0.5   | 94.51  | 94.85  | 89.66    | 78.13    | 84.21  | 94.85       | 78.40   | 87.56  |
> | 0.6   | 94.51  | 94.98  | 89.20    | 78.37    | 83.33  | 95.16       | 78.27   | 86.98  |
> | 0.7   | 93.98  | 94.98  | 88.61    | 78.01    | 82.75  | 95.31       | 76.44   | 86.45  |
> | 0.8   | 93.62  | 93.66  | 87.74    | 78.37    | 82.98  | 95.31       | 76.96   | 87.40  |
> | 0.9   | 93.20  | 92.34  | 86.72    | 78.01    | 83.26  | 94.77       | 76.18   | 87.19  |
> | 1     | 92.78  | 90.76  | 85.73    | 78.37    | 82.10  | 94.62       | 76.57   | 85.92  |
>
> We can observe that the optimal $\alpha$ resides within the (0, 1) which indicates that learning node representations from different feature spaces can effectively enhance the model performance. Notably, peak model accuracy occurs when $\alpha$ falls in (0,0.5], which may reveal that the topology information is more important than the attribute information in the node classification task.
>
> We will add the completed results and the corresponding discussions in the revised version.
>
> **Q4**. In A.4, the authors provide the results of SwapGT with different token sampling size. I have found that sparse splitting prefers larger sampling size. Can you provide some insightful discussions for this phenomenon?
>
> **A4**. Thank you for your helpful suggestions. The reason could be that the model faces more serious data sparsity issue under the sparse splitting. In this situation, a larger sampling size can bring more node tokens, which can alleviate the impact of data sparsity on the model performance. We will add the above discussions in the revised version.
>
> **Q5**. What is the value of the swapping probability $p$ in practice?
>
> **A5**. Thank you for your questions. In practice, we set $p$ as 0.5 in all experiments. We will clarify the setting of $p$ in the revised version.

---

> > ### Comment · Reviewer_n87V · 2025-08-02
> >
> > I thank the authors for their rebuttal. Based on the additional experimental results and discussions, I decided to keep my score and with the hope that the authors can improve their next version according to the above discussions.

---

> > > ### Author Response · Authors · 2025-08-03
> > >
> > > We are very grateful for your positive feedback and will carefully prepare the final version of our paper based on the experimental results and discussions provided in the rebuttal.

---

### Official Review · Reviewer_yYpa · 2025-07-04

**Clarity:** 3
**Significance:** 3
**Originality:** 3
**Rating:** 3
**Confidence:** 4

**Summary:**

This paper identifies a key limitation in existing tokenized Graph Transformers (GTs), where the token sequences for a target node are generated only from its immediate (1-hop) neighbors on a pre-constructed similarity graph. The authors argue this restricts the diversity of information available to the Transformer, limiting its performance, especially in sparse data settings. To address this, they propose SwapGT, a novel tokenized GT framework.

The core contribution is a token swapping operation, a data augmentation technique that creates multiple, diverse token sequences for each node by iteratively swapping tokens with those from their neighbors' token sets, effectively sampling from multi-hop neighborhoods on the similarity graph. Extensive experiments show that SwapGT consistently outperforms a wide range of GNN and GT baselines, particularly in sparse training scenarios.

**Questions:**

- The entire method depends on the initial k-NN graph. How sensitive is the model's performance to the choice of k in this initial step? Does a poor choice of k disproportionately harm SwapGT compared to a standard 1-hop GT?

- The token swapping operation is presented as the best way to leverage multi-hop information. How does this strategy compare, both conceptually and empirically, to other established multi-hop sampling techniques?

**Ethical Concerns:**

["NO or VERY MINOR ethics concerns only"]

**Limitations:**

yes

**Paper Formatting Concerns:**

No concerns.

**Quality:**

3

**Strengths And Weaknesses:**

Strengths
-  The paper does an excellent job of identifying a concrete and sensible limitation in the current design of tokenized GTs—the restricted sampling space for token generation. The framing of this process as a 1-hop neighborhood selection on a k-NN graph is insightful and provides a strong, clear motivation for the proposed solution.

- The proposed token swapping operation is an original and well-designed solution to the identified problem. It serves as an elegant, graph-specific data augmentation technique that expands the model's view to multi-hop semantic neighbors in a controlled fashion.

- The quality of the proposed method is demonstrated through rigorous experiments on eight benchmark datasets, including both homophilous and heterophilous graphs. The inclusion of standard deviation in the result tables further strengthens the reliability of these claims.

- The paper is well-written, clearly structured, and easy to follow. The overall framework is well-illustrated in Figure 2, and the token swapping algorithm is precisely detailed in Algorithm 1. The authors provide thorough ablation studies for each component of their method and include detailed experimental settings in the appendix, which aids in understanding and reproducibility

Weaknesses
- The method introduces several new hyperparameters, including the swapping times t, swapping probability p, number of augmented sequences s, and the fusion coefficient α. The parameter analysis in Figures 5 and 6 shows that model performance is sensitive to
t and s, and their optimal values vary across datasets. This reliance on careful tuning might limit the method's practical, out-of-the-box applicability.

- The paper includes a probabilistic analysis in Fact 1, showing that the swapping mechanism preferentially samples lower-hop neighbors on the k-NN graph. While this is a nice sanity check to show the process isn't completely random, it is a very light theoretical contribution. It does not provide any formal guarantees on the quality of the learned representations or why this specific sampling strategy leads to better generalization compared to others.

- The study on token generation strategies in Section 5.5 compares SwapGT against two "naive" variants: using a single, longer sequence (SwapGT-L) and randomly sampling from a larger pool (SwapGT-R). While SwapGT outperforms them, these baselines were designed by the authors for the ablation. The study would be more compelling if it compared the token swapping strategy against other established multi-hop or graph augmentation techniques from the literature, adapted to the tokenized GT context.

---

> ### Author Rebuttal · Authors · 2025-07-31
>
> Thank you for detailed comments. We provide the following responses:
>
>
> **Q1**. The entire method depends on the initial $k$-NN graph. How sensitive is the model's performance to the choice of $k$ in this initial step? Does a poor choice of $k$ disproportionately harm SwapGT compared to a standard 1-hop GT?
>
> **A1**. Thank you for your insightful questions.
> First, regarding the choice of $k$ in the $k$-NN graph: as noted in Lines 149–151, $k$ is indeed set equal to the sampling size of token sequences. In our original submission, we conducted extensive experiments to investigate how varying this token sampling size impacts model performance across datasets. Our findings consistently show that the optimal sampling size falls within the range of $k$=10 for nearly all datasets tested. This suggests that simply enlarging the length of token sequences, whether through dense or sparse splitting strategies, does not lead to performance improvements in most cases. We hypothesize this is because a larger sampling size increases the probability of introducing irrelevant or noisy nodes as tokens, ultimately degrading model performance. To compare the performance of SwapGT with the worst $k$ and the standard 1-hop GT, we evaluate the performance of standard 1-hop GT on the same sampling size of token sequences. The results are as follows:
>
> | Dense | Photo |  ACM  | Computer | Citeseer | Blogcatalog |  UAI  | Flickr | wikics |
> |:-----:|:-----:|:-----:|:--------:|:--------:|:-----------:|:-----:|:------:|:------:|
> | 1-hop | 94.59 | 93.08 |   90.63  |   75.56  |    93.87    | 75.31 |  85.27 |  82.09 |
> | Worst | 94.71 | 93.26 |   91.06  |   76.20  |    94.16    | 75.65 |  85.34 |  82.30 |
>
> | Sparse | Photo |  ACM  | Computer | Citeseer | Blogcatalog |  UAI  | Flickr | wikics |
> |:------:|:-----:|:-----:|:--------:|:--------:|:-----------:|:-----:|:------:|:------:|
> |  1-hop | 90.80 | 86.76 |   85.55  |   62.89  |    86.13    | 60.56 |  67.61 |  75.46 |
> |  Worst | 91.38 | 87.43 |   86.93  |   63.07  |    86.45    | 61.54 |  68.02 |  75.72 |
>
> From these results, we can clearly observe that SwapGT maintains superior performance over the standard 1-hop GT even in worst-case k scenarios. This is because that the token swapping operation can introduce more informative node tokens for node representation learning.
>
> We will add the above experimental results as well as the corresponding discussions in the revised version. Thank you.
>
> **Q2**. The token swapping operation is presented as the best way to leverage multi-hop information. How does this strategy compare, both conceptually and empirically, to other established multi-hop sampling techniques?
>
> **A2**. Thank you for your insightful questions. First, existing multi-hop node sampling-based strategy depend on the calculation of similarity scores between nodes. They only select the nodes with top-$k$ similarity as node tokens, which naturally ignores the semantic relevance between nodes. In contrast, our proposed token swapping operation can fully utilize the nodes’ semantic relevance to generate various token sequences, resulting in more informative token sequence and further leading to the performance improvement.
> Then, we compare the proposed method with a representative multi-hop node sampling-based tokenized graph Transformer called ANS-GT [1] which leverages random walk-based strategies to sample nodes from multi-hop neighborhoods. The results are as follows:
>
>
> |  Dense | Photo |  ACM  | Computer | Citeseer | Blogcatalog |  UAI  | Flickr | wikics |
> |:------:|:-----:|:-----:|:--------:|:--------:|:-----------:|:-----:|:------:|:------:|
> | ANS-GT | 94.88 | 93.92 |   89.58  |   77.54  |    91.93    | 74.16 |  85.94 |  82.53 |
> | SwapGT | 95.92 | 94.98 |   91.73  |   78.49  |    95.93    | 79.06 |  87.56 |  84.52 |
>
> | Sparse | Photo |  ACM  | Computer | Citeseer | Blogcatalog |  UAI  | Flickr | wikics |
> |:------:|:-----:|:-----:|:--------:|:--------:|:-----------:|:-----:|:------:|:------:|
> | ANS-GT | 90.42 | 88.58 |   84.95  |   62.94  |    78.26    | 55.85 |  64.42 |  75.72 |
> | SwapGT | 92.93 | 90.92 |   88.14  |   69.91  |    88.11    | 63.96 |  72.16 |  78.11 |
>
> We can observe that SwapGT consistently beats ANS-GT on all datasets under different splitting strategies, which indicates the effectiveness of token swapping operation in SwapGT for node classification, compared to representative sampling-based tokenized GT.
>
> We will add the above results and discussions in the revised version. Thank you.
>
> [1] Zaixi Zhang, et al. Hierarchical graph transformer with adaptive node sampling. NeurIPS 2022.

---

> ### Comment · Area_Chair_hcDr · 2025-08-05
> **Discussion Reminder**
>
> Dear Reviewer,
>
> The authors have submitted a rebuttal. Please take a moment to read it and engage in discussion with the authors if necessary.
>
> Best regards,
> AC

---

> ### Author Response · Authors · 2025-08-07
>
> We sincerely thank Reviewer **yYpa** for recognizing our work. As we are currently within the reviewer-author discussion period, if you have any further questions or concerns regarding our work, we warmly welcome your feedback. We will do our utmost to address and clarify them. Once again, we truly appreciate your valuable time in reviewing and engaging in the discussion.

---

### Note · Authors · 2025-08-12

During the rebuttal phase, we clarified issues regarding the innovation, motivation, and effectiveness of our work, engaged in active discussions with three reviewers, and ultimately gained their recognitions. Unfortunately, Reviewer **yYpa** did not participate in the discussion. While this reviewer acknowledges the innovation of our method and the contribution of the paper, the main concern focused on the ablation studies of the method. In the rebuttal, we have fully validated the effectiveness of the designs in our proposed method through additional experiments according to the reviewer's comments.

Thank you to all reviewers and the AC for your time!

---

### Decision · Program_Chairs · 2025-09-17

**Decision:**

Accept (poster)

**Comment:**

In this paper, the authors study node tokenized graph Transformers (GTs) for node classification. The authors argue that existing GTs only consider first-order neighbors on the similarity graphs, and propose SwapGT, adopting swapping operations to generate more informative token sequences. It also adopts a center alignment loss to constrain the representation learning. Experiments show the effectiveness of SwapGT for node classification.

The paper received mixed comments. The reviewers agree that the paper is clearly written, well-structured, and motivated by a concrete limitation in existing tokenized GTs. Initially, several concerns regarding technical novelty and insufficient experiments have been raised. During rebuttal, the authors have provided more results and explanations, which successfully address the concerns of three reviewers. The only remaining negative reviewer did not engage in discussions all along. I have briefly checked the paper and review comments. Despite some potential weaknesses, such as the technical analyses not being very deep, its core idea is novel and the empirical validation is extensive, so the paper makes a meaningful contribution to advancing tokenized GTs. I therefore recommend acceptance.